# Residential Environment Assessment by Older Adults in Nursing Homes during COVID-19 Outbreak

**DOI:** 10.3390/ijerph192316354

**Published:** 2022-12-06

**Authors:** Fermina Rojo-Perez, Vicente Rodriguez-Rodriguez, Gloria Fernandez-Mayoralas, Diego Sánchez-González, Carmen Perez de Arenaza Escribano, Jose-Manuel Rojo-Abuin, Maria João Forjaz, María-Ángeles Molina-Martínez, Carmen Rodriguez-Blazquez

**Affiliations:** 1Grupo de Investigacion sobre Envejecimiento (GIE), IEGD, CSIC, 28037 Madrid, Spain; 2Department of Geography, National Distance Education University (UNED), 28040 Madrid, Spain; 3Unidad de Analisis Estadistico (UAE), CCHS, CSIC, 28037 Madrid, Spain; 4National Centre of Epidemiology and Health Service Research Network on Chronic Diseases (REDISSEC) and Research Network on Chronicity, Primary Care and Health Promotion (RICAPPS), Carlos III Institute of Health, 28029 Madrid, Spain; 5Department of Personality, Evaluation and Psychological Treatment, Faculty of Psychology, National Distance Education University (UNED), 28040 Madrid, Spain; 6National Centre of Epidemiology and Network Centre for Biomedical Research in Neurodegenerative Diseases (CIBERNED), Carlos III Institute of Health, 28029 Madrid, Spain

**Keywords:** COVID-19, older adults, long-term care settings, residential assessment, Madrid region, Spain

## Abstract

The most vulnerable residential settings during the COVID-19 pandemic were older adult’s nursing homes, which experienced high rates of incidence and death from this cause. This paper aims to ascertain how institutionalized older people assessed their residential environment during the pandemic and to examine the differences according to personal and contextual characteristics. The COVID-19 Nursing Homes Survey (Madrid region, Spain) was used. The residential environment assessment scale (EVAER) and personal and contextual characteristics were selected. Descriptive and multivariate statistical analysis were applied. The sample consisted of 447 people (mean age = 83.8, 63.1% = women, 50.8% = widowed, 40% = less than primary studies). Four residential assessment subscales (relationships, mobility, residential aspects, privacy space) and three clusters according to residential rating (medium-high with everything = 71.5% of cases, low with mobility = 15.4%, low with everything = 13.1%) were obtained. The logistic regression models for each cluster category showed to be statistically significant. Showing a positive affect (OR = 1.08), fear of COVID-19 (OR = 1.06), high quality of life (OR = 1.05), not having suspicion of depression (OR = 0.75) and performing volunteer activities (OR = 3.67) were associated with the largest cluster. It is concluded that a better residential evaluation was related to more favourable personal and contextual conditions. These results can help in the design of nursing homes for older adults in need of accommodation and care to facilitate an age-friendly environment.

## 1. Introduction

In Spain in 2013, out of just under 450,000 people in collective establishments, around 60.9% were older adults [1]. More recent estimates of older adults have reported a total of 384,251 bed places in 2020, with an occupancy level of 75–80% [2]. Of the total number of places, 13.5% were located in the Madrid region [3].

At older ages, living in a nursing home is a significant change in one’s way and quality of life. Among the reasons given in the literature for moving into nursing homes are personal and family factors, such as advanced age, having few children [4], inadequate housing status or the presence of accessibility barriers in the residential environment, family pressures [5], cognitive impairment, not being married, and being diagnosed with dementia [6], while residential facilities can compensate for loss of capacity and independence [7]. One of the most commonly noted rationales in research relates to health conditions, including health status and physical and cognitive impairment, as the main determinants for entering residential care [4,8]. Thus, the desire to receive long-term care in a suitable environment close to the previous place of residence, such as that provided in nursing homes, is a key reason for entering there [9].

A decline in quality of life has been observed with access to institutionalization [10]. Thus, recent studies have emphasized the need to increase the perception of quality of life, especially through interventions aimed at preventing depression and improving residents’ self-rated health [11]. Compared to the strategy of remaining in the usual family home, institutionalization could entail a higher risk of vulnerability, as older adults in these centres are often affected by various conditions of ill health, such as comorbidity, functional impairment, or depression, as well as loneliness [12].

This background provides a basis for the study of the older adult population in residential homes during the COVID-19 period. This pandemic crisis has resulted in many scientific studies being published [13], most of them of an epidemiological and clinical nature, and to a lesser extent of a social background. Some of these publications are aimed at understanding the situation of the pandemic in collective residential settings, such as nursing homes.

The effects of the pandemic have been felt across all socio-demographic groups, but the impacts have been exceptional in the nursing-home population, which is affected by increased frailty and poor health, as well as cognitive and behavioural deficiencies [14], in addition to other nursing-home-related organizational and management aspects [15]. This has occurred in both Europe and North America during the different epidemic waves, albeit with variations depending on the incidence of the geographical areas of location [16].

The impact of this disease has been felt by residents, workers, and family members [17]. Giri et al. have pointed out several factors, such as the characteristics of the residents (comorbidity, dementia) and of the homes (size, high occupancy, location in urban areas, for-profit facilities, rapid spread of the disease), as well as other circumstances, such as the uniqueness of the disease itself, the downsizing of staffing levels and staff burnout, the non-availability of care technology, the reduction in non-pharmacological measures (family and social connections, physical and cognitive stimulation, phone or video calls, and less mealtime conversation or fewer relationships with other residents due to being confined in one’s own room), and other confinement-related effects on residents [18]. The conceptual framework of a scoping review, designed to understand COVID-19 transmission factors in long-term care homes, included criteria regarding accessibility of services, quality of services, value for money, and quality of care [19].

Facilities with larger numbers of beds or higher staffing levels reported higher disease incidence during all waves [20,21], although higher staffing levels were associated with fewer deaths [22]. Very small or very large nursing homes had high mortality rates, suggesting that intermediate-sized facilities (between 30 and 70 beds) would be optimal for preventing infection and other related negative consequences [23]. Community-dwelling studies have also reported a higher impact of disease transmission in overcrowded dwellings, which generate unavoidable interactions and limit isolation capacity [24].

To prevent and contain the incidence of the disease, control measures were established in residential facilities for residents, workers, and family members [25,26,27,28,29]. In essence, the rules consisted of confining residents to appropriate spaces, restricting visitors, establishing hygiene measures, the cleaning and disinfection of spaces and devices, control policies, and the management of infected persons (e.g., residents and workers) [15].

Faced with COVID-19′s impacts and the difficulties brought about by confinement, numerous scientific publications have emphasized different issues, such as the importance of addressing the need for telemedicine-based consultations with health professionals to allow remote assessment by specialists [30], but also virtual relationships with family or friends [31,32]. Despite worker and staff efforts to help residents use electronic devices to stay in touch with their relatives [31,33], the still-poor provision of this technology, the fact that it cannot be adapted to residents’ needs, and this demographic group’s digital divide have hindered communication during the pandemic [34], which has also led to an increased risk of perceived isolation and loneliness [35]. In response to this isolation, the use of advanced technologies, as well as the maintenance of virtual intergenerational contacts, is proposed to improve social interaction [36].

Various aspects of COVID-19′s influence have been noted in [37], such as those on livelihoods [38]; socio-economic issues [39,40]; business, consumption, employment, market [41]; sustainable development goals [42]; education [43,44]; tourism and travel [41,45,46]; work, workers, remote work [45,47]; environmental implications [48]; and urban agglomerations [49], or the exodus towards suburban areas of higher environmental quality [50], the latter process having begun before the pandemic and gathered pace during COVID-19 [45].

Many studies in a residential context have focused on the built environment [51,52], the attributes of urban space and green areas [53], the redesign of a post-pandemic residential environment for quality-of-life improvement [54], the restriction of routine physical activities in the residential environment [55], the concentration of activities and services in the built environment [56] and housing [54], mental health and the built environment [57], and barriers to active and healthy ageing [58].

This health crisis must be used to address solutions and ideas in designing the residential environment to provide quality of life for residents [37], because ageing cannot be slowed down; however, as Wang points out, controlling pandemic outbreaks (avoiding entry, preventing spread, and controlling infection and disease) involves redesigning the environmental setting on different spatial levels (of the nursing home, of the building and residents’ rooms, and of recreational areas) [59].

In relation to experiences in the collective residential environment, such as homes for older adults, studies have centred on issues relating to the decision to enter a nursing home, the perception and experience of the nursing home before the pandemic, and the changes perceived during the pandemic [60]. Additionally, the perceptions and feelings of older people living in the community and in sheltered housing have been analysed from a comparative approach [61]. However, there is a gap regarding the study of and knowledge about the built space, social space, and the experiences of residents in nursing homes during the COVID-19 pandemic. Thus, this paper focuses on the perception of residential environment elements regarding physical features and amenities, social relations, accessibility and mobility, and the availability of space for personal use. The goals of this paper are: (i) to ascertain how older people assessed their residential environment during the COVID-19 outbreak in long-term settings; (ii) to examine the structure and reliability of the scale of the residential environment assessment; (iii) to group subjects into clusters of similar characteristics according to residential assessment; iv) to find out the personal and contextual facts that condition the position of the subjects in each of the residential assessment clusters. It is assumed that better personal and contextual conditions will be associated with more favourable perceptions of the residential environment.

## 2. Materials and Methods

### 2.1. Data Source and Selected Variables

This study has used the survey of the research project *Nursing Homes and COVID-19: Environments of Older People as Protectors in Health-Emergency Situations (COVID-19*), carried out between June and October 2021 in the region of Madrid, Spain. The technical characteristics of the survey (sampling, significance), the selection of participating nursing homes (according to typology, size, and location), the distribution of cases per nursing home, information regarding the contact with nursing homes and participants, the fieldwork, the ethical declaration, the informed consent, thematic blocks of the survey and content, and the basic results, can be found in Rodríguez-Rodríguez et al. [62]. The number of cases was 447. The sampling error was ±4.8%, for the estimation of percentages under the assumption of maximum variability (*p* = q = 0.5) and a confidence level of 95%.

The following variables have been selected for the purposes of this work:

(a) Perception of the residential environment: the 10-item residential environment assessment scale (EVAER, acronym based on its name in Spanish: “Escala de VAloración del Entorno Residencial”) was employed. This scale was designed according to a broad understanding of the residential environment, which includes both physical and social aspects [63] that influence residential satisfaction and quality of life in old age [64,65]. It consists of 10 items measured on a Likert scale of 5 points, assessing aspects of the nursing home and daily life (very good, good, fair, poor, very poor), grouped in 4 domains: (i) relationships (with workers, residents, family, and friends); (ii) aspects regarding facilities and resources, the perception of safety and management during the pandemic; (iii) mobility within and outside the centre; and (iv) availability of single or shared rooms and other spaces for personal use. Higher values indicate a better assessment. In addition, the use of space-related variables (type of room where most of the time was spent during the pandemic) and the means used to maintain relations with family or friends during the pandemic lockdown were also used in this study;

(b) Socio-demographic characteristics: age, sex, marital status, educational level, number of children;

(c) Personal situation with regard to the pandemic: self-perception of having had the disease, worry about the pandemic, the Fear-of-COVID-19 Scale (FCV-19S) [66] (Cronbach’s alpha = 0.94), and preventive measures taken to cope with the pandemic (up to 7 measures surveyed);

(d) Objective physical health, provided by the nursing home (number of illnesses and medications taken for each resident surveyed). Self-perception of health was also used (higher values indicate better self-perception). The Geriatric Depression Scale abbreviated to 5 items (GDS-5) [67] was used as a measure of mental health (Cronbach’s alpha = 0.207), and a score of 2 or more indicates suspicion of or confirmed depression;

(e) Scales for feelings and coping: (i) the Positive and Negative Affect Schedule (PANAS) [68]—in this study, positive and negative affect scores were calculated (Cronbach alpha = 0.798 and 0.808, respectively) to obtain the measure of balance to be used in the analysis; (ii) the Brief Resilient Cope Scale (BRCS) [69] (Cronbach’s alpha = 0.91); and (iii) the perception of loneliness as measured by the frequency of feeling lonely.

(f) The performance of leisure and participation activities was assessed by classifying subjects into 4 groups obtained by cluster analysis, based on the frequency of performance of up to 8 surveyed activities: group 1—use of ICT devices, such as a computer, tablets, mobiles, etc., to look for information, chat, etc., on the internet (3.4% cases); group 2—rewarding activities, such as handicrafts, doing things for others/volunteering (29.3%); group 3—inactivity, which includes people with very low participation in all activities (62.2%); and group 4—religious practice, beside a very low participation in all activities (5.1%). Applying discriminant analysis to the activity items showed that 94.9% of the original grouped cases were correctly classified [70];

(g) Quality of life in older adult nursing homes based on the abbreviated FUMAT scale, which contains 24 items grouped in 8 dimensions—1: emotional well-being, 2: interpersonal relationships, 3: material well-being, 4: personal development, 5: physical well-being, 6: self-determination, 7: social inclusion, and 8: rights [71]. Cronbach’s alpha for the global scale was 0.79. Other variables related to well-being and quality of life were life satisfaction and an assessment of the stay in the nursing home, both of which involved a comparative perspective, i.e., before and during the COVID-19 pandemic. The self-perception of old age was measured with the five-item Attitude Toward Own Aging subscale (ATOA) of the Philadelphia Geriatric Center Morale Scale (PGCMS) [72,73], with a Kuder–Richardson reliability coefficient KR20 = 0.64. For all variables in this group, higher values indicate a better position on quality of life or self-perception of ageing.

### 2.2. Statistical Analysis

In order to meet the study objectives, the analytical process has been divided into several phases:

- Firstly, descriptive statistical analysis was used to examine residents’ ratings of different aspects of the residential environment (EVAER scale).

- Exploratory factor analysis (EFA) and Cronbach’s alpha techniques were used to ascertain the properties of the EVAER scale items and to resolve on their clustering.

- A classification of homogeneous groups of subjects according to their residential assessment was obtained by cluster analysis, and discriminant analysis was applied to validate the cluster classification.

- Binary logistic regression analysis was applied to each of the residential assessment clusters, in order to ascertain what determines a person’s position in each regression model factor category. A stepwise selection method (forward-selection conditional) was used. Of all the personal and contextual variables selected in the study, those that were significant were used by applying one-way ANOVA, with scale and ordinal variables, and chi-square test with nominal variables.

## 3. Results

### 3.1. Overall Residential Assessment Characteristics

The sample consisted of 447 people, 63.1% of whom were women, and the mean age was 83.8 years. The general characteristics of the participants, as well as the rest of the dimensions investigated in the survey, can be found in Rodríguez-Rodríguez et al. [62].

The residents’ assessment of the residential environment is relatively high (Table 1). The elements perceived as good/very good by more than 9 out of 10 people were the management of the nursing home during the pandemic period, self-perception of safety in the nursing home, relationships with workers and family members, availability of a space for privacy (having a single or shared room with spouse/family or other spaces for personal use), and the characteristics of the residential environment in terms of resources and facilities. In a smaller proportion of cases, respondents also highlighted relationships with friends and other residents, and lastly, indoor accessibility, i.e., the ease of moving around inside the nursing home, as well as the possibility of going outside the nursing home for different activities (such as walking, shopping, leisure, visits, etc.).

### 3.2. The Assessment of Residential Environment Scale Properties

The internal consistency or reliability through the Cronbach’s alpha coefficient for all 10 items of the EVAER scale (0.84), the Kaiser–Meyer–Olkin (KMO) (0.823) and Bartlett’s Test (𝜒2 = 976.520, df = 45, *p* < 0.001) indicated that the scale was acceptable for factor analysis. Thus, the EFA was applied with all items of this scale (Appendix A). The results showed high communalities and four factors were obtained, which, together, explained 72.2% of the accumulated variance, and were named as: 1: Relationships (with workers, residents, family, and friends); 2: Nursing-home aspects (facilities and resources, perception of safety, management of the nursing home during the pandemic); 3: Mobility (mobility outside and inside the nursing home); and finally, factor 4: Privacy space (availability of an individual space or other personal spaces). The results supported the construction of four residential assessment subscales, with an adequate Cronbach’s alpha to accept the reliability hypothesis: relationships (0.77), nursing-home aspects (0.72), mobility (0.74) and privacy space (no Cronbach’s alpha for this subscale based on a single variable). An aggregate score of items was used as an indicator of each subscale, with a higher score meaning a better assessment.

### 3.3. Grouping of Subjects According to Residential Assessment

Cluster analysis was applied using the k-means algorithm to identify relatively homogeneous groups of subjects according to the residential environment assessment. The input variables were the standardized (mean = 0.0; standard deviation = 1.0) scores of the four EVAER subscales. Three clusters, mutually homogeneous and different from the rest of the clusters, were obtained: (i) medium-high rating of all residential aspects, with 71.5% of the subjects; (ii) low rating linked to mobility (15.4%); and (iii) low rating of all residential aspects (13.1%) (Appendix A). This grouping of subjects was validated by discriminant analysis, so that 98.3% of the original cases were correctly classified. Figure 1 shows the clusters in the discriminant function space, visually presenting the separation of the clusters.

### 3.4. Explaining Membership of Each Residential Value Cluster

The variables used in each regression model were the ones that were significant in the bivariate analysis (Appendix A). Educational level, coronavirus status, concern about the pandemic, number of diagnosed illnesses, and medications taken were not significant.

#### 3.4.1. Prediction of Medium-High Rating with All Aspects of the Residential Environment

In the first logistic regression model, the response variable that was statistically significant (χ^2^ = 88.375; *p* < 0.001) was the cluster of medium-high rating of all aspects of the residential environment (Table 2). The explained variation in the dependent variable ranged from 21.5% (Cox and Snell R square) and 31.1% (Nagelkerke R square). The Hosmer and Lemeshow test with a non-significant chi-square (χ^2^ = 5.592; *p* = 0.693) indicated that the data fit the model well. The model correctly classified 77.5% of overall cases within this cluster.

The variables that were significant (*p* < 0.05) in the model and that achieved an OR (odds ratio) slightly above 1.0 were expressing a positive affect (OR = 1.08), declaring fear of COVID-19 (OR = 1.06), and higher quality of life (OR = 1.04). Similarly, lower levels of depression were associated with a higher likelihood of having a better rating of all aspects of the residential environment (OR = 0.75). In relation to the frequency of leisure activities, going from being inactive to belonging to the group of subjects who engage in rewarding activities (such as volunteering or doing things for others) increased the residential rating by 267.3% (OR = 3.67) while religious practice decreased it by 68.1% (OR = 0.32).

On the other hand, the likelihood of having a medium-high residential rating decreased when going from having a single room to sharing it with residents other than a spouse or partner or another family member (OR = 0.46). During the COVID-19 lockdown, going from spending most of the time in the room to limited or unrestricted use of other spaces in the nursing home decreased the likelihood of belonging to this group (OR = 0.46).

#### 3.4.2. Prediction of Low Rating with Mobility in the Residential Environment

In the second model, the response variable that was statistically significant (χ^2^ = 66.298; *p* < 0.001) was the cluster of low assessments with respect to mobility (inside and outside the nursing home) (Table 3). The explained variation in the dependent variable ranged from 16.6% (Cox and Snell R square) and 29.1% (Nagelkerke R square), the data fit the model (Hosmer and Lemeshow test: χ^2^ = 6.681; *p* = 0.463) and 87.1% of cases were correctly predicted.

There were five resulting variables in the model, four of which were directly associated with an increased likelihood of having a low rating for mobility inside and outside the nursing home. Thus, those who reported using non-owned electronic devices to maintain contact with family and friends during the pandemic were almost four times more likely to have a low mobility rating (OR = 3.75) than those who reported using their own devices. Women were 3.25 times more likely to have a low mobility score. A high self-perception of stay in the nursing home in comparative perspective (before and during the COVID-19 pandemic period) raised the likelihood of having a low mobility rating (OR = 2.82). Increased depression was associated with an increased likelihood of low mobility assessment (OR = 1.42).

In relation to the frequency of leisure activities, moving from the inactive group to the rewarding group (helping others, volunteering) decreased the likelihood of belonging to this cluster (OR = 0.17), while the remaining categories of activity were not significant.

#### 3.4.3. Prediction of Low Rating wih All Aspects of the Residential Environment

The response variable of the third model was the cluster of low rating with all aspects of the residential environment, and this was statistically significant (χ^2^ = 84.740; *p* < 0.001). (Table 4). The explained variation in the dependent variable ranged from 20.7% (Cox and Snell R square) and 39.4% (Nagelkerke R square) and the data fit the model (Hosmer and Lemeshow test: χ^2^ = 6.760; *p* = 0.563). With the independent variables added, the model correctly classified 91.0% of cases overall.

Six exposure variables were entered into the model. The likelihood of giving all residential environment elements a low rating is higher among those who reported having made limited or unrestricted use of other spaces in the nursing home than their own room (OR = 7.01), or being single (OR = 6.29) or widowed (OR = 4.06). Life satisfaction in comparative perspective (before and during the pandemic) was also associated with this cluster (OR = 3.33). Conversely, a decrease in quality of life, affect, and self-perception of the stay in the nursing home in comparative perspective was associated with a higher likelihood of belonging to this cluster (OR = 0.93, 0.89, 0.22, respectively).

## 4. Discussion

This study has examined the profiles of institutionalized older people according to their experience and assessment of the residential environment during the COVID-19 period, as well as the relationship of these profiles with personal and contextual conditions. The setting studied is older adults’ nursing homes in the region of Madrid, Spain. The analysis is part of a broader research project aimed at ascertaining the environmental and psychosocial factors affecting COVID-19′s incidence on older people’s perception of the nursing homes where they live. A semi-structured survey was designed and implemented to collect information that included several environmental, social, physical, and emotional health dimensions.

The scientific literature reviewed has highlighted the high incidence of the disease and associated mortality in the older adult population in the early stages of the disease, with residents in nursing homes being the most highly affected [36,74,75], as has occurred in Spain [76,77,78]. This population group, due to its own socio-demographic conditions and age-associated chronic comorbidity and functioning, was the most vulnerable to SARS-CoV-2 infection, hospitalization, and death [61,79,80,81], with women being the group most affected by mortality [82]. This has been compounded by the rapid spread and high risk of infection owing to person-to-person contact between workers and residents [83]. In some cases, this has led to the establishment of COVID-19 units to care for affected patients [84].

The residential environment acts as a container within which life develops. This is why the friendliness of the built space can help its residents to maintain or even improve their quality of life [85]. Thus, action must be taken in this environment in the face of possible future pandemics as “residential buildings are crucial for the health of the population, as they determine social well-being. Homes had critical importance during lockdown periods during which people were required to stay home for infection spread prevention. The nature of this home quarantine experience differed significantly from person to person. This invokes a substantial rethinking of housing to prepare humanity for future possible disease outbreaks” [37] (see page 15).

This paper fills a gap in the context of nursing homes during COVID-19. Other authors have analysed the experiences of older adults in family and sheltered housing during the outbreak [61,86]. The purpose of this study was addressed by designing the ten-item EVAER scale, covering several residential environment dimensions, in order to ascertain how residents themselves assess aspects of daily life in nursing homes during the pandemic. The factor structure and reliability analysis of this instrument revealed four dimensions (relationships, resources, mobility, privacy space). This short scale can be employed to obtain survey information in multidimensional studies. The use of other instruments with a wealth of content, such as the Perceived Residential Environment Quality Index (PREQI) [87,88], was discarded, as they transcend the objectives of this study both from the thematic point of view and in terms of the length of either the long or abbreviated version of the instrument [89,90], in addition to the fact that this index is designed to determine the quality of the residential environment for the general population.

It was assumed that better personal and contextual conditions were associated with the profiles of subjects with a more positive assessment, and this assumption has been verified in the results obtained. Of the three groups of subjects identified, the largest is the one that stated a medium-high rating for all the residential environment elements, which included more than 7 out of 10 people. This may reflect the level of satisfaction, as an overall indicator of quality of life, of older adults in residential care. Indeed, the results revealed that two-thirds of older adults were frequently or always satisfied with their current life in terms of expectations and lifestyle, and that a quarter of the residents reported that the stay in the nursing home was better than before the pandemic [62]. This is in line with other pre-pandemic studies, which, while noting a decline in well-being after admission to the facility, around 6 months later show scores reverting to pre-admission status [91]. In fact, global quality of life has been shown to be a predictor in two of the models obtained (medium-high and low rating with all aspects of the residential environment). However, while in the first one, quality of life was directly associated, in the second one it was negatively associated. This is a result that has already been observed in the case of Istanbul (Turkey), where the quality of life of the institutionalized older adult population was above the mean values measured from the WHOQOL-BREF Turkish Version scale and the dimensions with the lowest values were the social relations and physical dimensions, while the least affected were the mental and environmental dimensions [92]. In our study, the global indicator of quality of life has been used to avoid increasing the already large volume of information used. Resilience may also underlie this level of satisfaction and adaptation. Although this factor has not been found to be a predictor in the models, resilience has been shown to be a moderate value in this study (mean: 15.8; minimum: 4; maximum: 20) [62], as has been found in other research, and this is perhaps in line with the relationship between both quality of life and resilience constructs [92].

Reporting fear of COVID-19 was associated with a better assessment of the residential environment as a whole. Fear reached a moderate mean value [93], similar to that found in studies in Turkey [92,94] or in the study of nurses in emergency services during COVID-19 in Peru [95]. According to other research, older adults have been the most affected by reporting a feeling of fear of COVID-19 in the first wave, compared to general population in Spain, Slovakia, and Slovenia, while in Italy it was higher, and this is related to a higher incidence of infection and consequent deaths [96].

The psychological and emotional effects of COVID-19 have been enormous [97] and this can lead to a deterioration of mental health [98]. Both the personal influence of having been infected by the virus, and the community influence of the effects of the measures designed and implemented to contain the infection, have manifested themselves in a variety of emotional impacts. Thus, in this study, mood and affect, based on the PANAS scale, were associated with model 1 (medium-high rating) directly and with model 3 (low rating) inversely. In other studies, affect was clustered with the different dimensions of fear of COVID-19, such that positive affect and fear were inversely related [99]. In other follow-up research on the emotional impact of the pandemic on the adult population in Spain, an increase in positive affect and a decrease in negative affect has been found [100], while older adults showed a lower level of psychological distress (measured through anxiety, anger, sadness, fear, and hope) [101], which was also shown in a study on adults in the US, in which older adults showed more positive affect and less negative affect [102]. From a comparative approach, the effects of isolation and loneliness have been shown to influence feelings of worry, stress, anxiety, fear, frustration, boredom, and depression in both older adults in residential care and those living alone in family housing [103]. In our study, maintaining positive emotions and attitudes has been shown to have a constructive effect on a better appreciation of the residential environment in institutionalized older adults.

As another psychological effect, depression was a significant factor in models 1 and 2, so that not suffering from depression was associated with medium-high ratings for all residential elements, while suffering from depression influenced the likelihood of a low mobility rating. Along with other aspects that deteriorate well-being and mental health, depression during COVID-19 has increased in older adults and in people with vulnerable health conditions [98]. In this regard, much research has shown high levels of depression, as in the case of older people in residential care in Israel [60], New Zealanders of European origin [104], in Malaysia [105], and in the US [106]. Depression has also been reported in the older-adult population in community-dwelling [107].

Regarding residential resources, having private rooms with quality amenities (bathroom, natural light, adequate size) is associated with quality of life and with improving infection control and having a space for isolation in case of suspected infection [108]. Additionally, avoiding overcrowding decreases the likelihood of COVID-19 outbreaks [109]. In this regard, the provision of a single room is relevant for privacy and intimacy, and staying in one’s own room during COVID-19 may have led to a feeling of safety from infection. Therefore, sharing a room on the one hand, and using the different areas of the nursing home on the other hand, decreased the likelihood of reporting a medium-high rating for all aspects of the nursing home (model 1). On the other hand, limited or unrestricted use of the nursing-home spaces influenced low residential ratings (model 3).

In relation to leisure activities, the period of old age is appropriate for participation in leisure activities tailored to each person’s circumstances, even for those not carried out in other periods of life. In fact, quality of life in old age is influenced precisely by active participation along with health and functioning [7]. Older adults conceive it under this prism in the context of active ageing [110,111]. However, the pandemic and lockdown experience have prompted a reduction in the frequency of participation in leisure activities [112], especially those with a social and active component performed outdoors [113], as well as the need to adjust leisure to become more passive [114]. In the context of long-term care facilities in the Netherlands, a process of (dis)continuation of activities has been observed through measures of organizing activities in different locations, in limited groups, and subject to an action guide [115]. Our study has shown that more than 6 out of 10 people were grouped in the inactivity cluster (very low participation in activity) [70]. Thus, the category of people involved in supporting and helping others was associated with a higher likelihood of belonging to the medium-high rating group for all aspects of the residential environment, which may indicate the emotional reward for participating in this type of activity.

The high risk of older adults being infected in nursing homes led to the establishment of restrictive measures both for leaving the nursing home and for receiving outside visitors. In this context, many centres facilitated remote communication between residents and relatives through electronic devices [31], not only through phone calls [33], albeit sometimes with little success [26] due to a lack of infrastructure as well as the digital divide. Our results indicated that the likelihood of having a low mobility rating was increased by having to use electronic devices owned by other residents or even staff for communicating with relatives or friends outside. Therefore, attention should be paid not only to the lack of skill in the use of these devices but also to the unavailability of them.

Despite the impact of being confined during COVID-19 on various aspects of nursing home life, other studies have found that the restrictive measures introduced in nursing homes during the pandemic have led residents to feel safe and secure in relation to the transmission of the virus, apart from feeling limited in their freedom and dependent on staff [116]. Safety and management in the nursing home were integral aspects of the EVAER scale, so it should be emphasized that participants are rating these items positively. In a qualitative Swedish study [116], respondents highlighted the aspects underlying the feeling of security, in the sense that living in an institution means that there is a body of staff to care for and support the resident; for its part, being confined has a dual effect, one of which is negative (loneliness, isolation), while the other is positive (security against the transmission of the virus). In this regard, the literature review by Tokazhanov et al. on the sustainability of residential buildings advocates the need to consider various requirements to provide health, safety, and comfort without harming the environment [37], although in nursing homes being confined to a room, as a physical safety measure to contain infection, has, on the contrary, had other impacts on the emotional health of the elderly [60].

Gender was only associated with the likelihood of having a low mobility rating (model 2), with females being the worst positioned. Among the restrictive measures, mobility was limited both within the nursing home and for going outside. In this context, the association of gender with low mobility scores could also be explained by a higher effect of chronic health conditions among women. In this study, women reported a higher number of diseases and medication intake than men (*p*-value ≤ 0.05). In the family-dwelling population, older women have been known to be more housebound than men [107], derived from their greater commitment to household and caregiving activities [117]. In relation to marital status, being single or widowed increased the likelihood of having a low rating with all residential environment elements (model 3). This could be related not only to being alone (single, widowed) but also to the feeling of loneliness; in fact, the results showed that single and widowed people feel equally or even more lonely than before the pandemic.

## 5. Concluding Remarks and Future Research

The friendliness of the residential environment, which increases the person–environment fit [118], must prioritise the habitability of spaces from an inclusive approach and considering all the dimensions established in the Age-Friendly Cities and Communities paradigm [119]. This is the only way to mitigate the negative impacts of pandemic periods in order to achieve the required levels of well-being during ageing. In this sense, the results obtained, especially regarding social inclusion and mobility restrictions, the deterioration of participation in outdoor activities, disconnection in family and social relationships, and the availability of private and other community spaces, can constitute a knowledge base to promote safe and well-being environments for their residents.

This study is not without its limitations. Although it responds to very specific objectives, it should be noted that not all the elements involved in the care of the institutionalized older adult population during the COVID-19 period have been considered. As highlighted in other publications from the pandemic period, both nursing-home workers and family members have been involved in caring for older adults, as well as in the possible transmission of infection [15,17,21,26,83,120,121]. Workers have reported COVID-19 effects related to emotional health and workload due to staff shortages, as well as concerns about severe isolation measures, death of residents, and fear of transmitting the virus to family members [122]. For their part, family members mentioned great concern about understaffing, as well as the absence of infected workers, limitations in communication between family members and caregivers on the one hand, and family members and residents on the other, due to restrictions and the digital divide [26]. This made it difficult for them to know how their relative was and the quantity and quality of care provided to the resident [123]. Thus, a future line of research to overcome this limitation and to ascertain the full extent of the situation in the studied environment intends to address the assessment and experience of the contexts involved (residents, workers, relatives) using a mixed methodology, both in nursing homes and in cohousing settings, as another type of transition between family housing and institutionalization.

Another line of future research focuses on the determinants of quality of life, addressing this construct from the global and also domain-specific approach, in order to ascertain which quality-of-life dimensions have been most affected by the pandemic, in line with other studies [92], and what the underlying determinants are.

On the issue of the sense of security expressed by the residents in the residential environment during the COVID-19 period, the determinants underlying this perception must be ascertained, and this must also be approached using qualitative methodological techniques that facilitate discourse analysis. Other studies have found that older people in institutions felt physically safe, but solitary confinement led them to state that the centre had become a prison, although in comparative perspective they also felt that other older people were perceived to be worse off than themselves [60]. As it is of relevance in all dimensions of life and old age, this issue of security must be addressed further, given that it is a pillar of active ageing [124], of the Age-Friendly Cities and Communities paradigm [119], and one that older adults in all residential contexts value to a relevant degree, especially those who are institutionalized [125].

These results suggest recommending the provision of personal/private spaces in residential settings for older adults, in order to maintain their privacy and to have an individual space in which to avoid contact with other residents in the event of infectious outbreaks, as has happened with COVID-19. They also point to the need to provide information and communication technologies as a means of relating to the outside world (family, friends, acquaintances, medical consultations, other consultations), to avoid feelings of isolation and loneliness, as well as telemedicine-based access to health consultations [34]. Providing residents with the knowledge and strategies for its use is also crucial. This is a fact that has greatly encouraged isolation among residents and disconnection with their families [33,35], despite the efforts of workers [31].

This and other similar studies would provide public policies with essential data for designing and implementing measures to improve the quality of life of older adults living in nursing homes. Nursing homes for older adults are, in fact, the home of all their residents, and their consolidation as such is not only a duty towards them, but also a right that is theirs as individuals.

## Figures and Tables

**Figure 1 ijerph-19-16354-f001:**
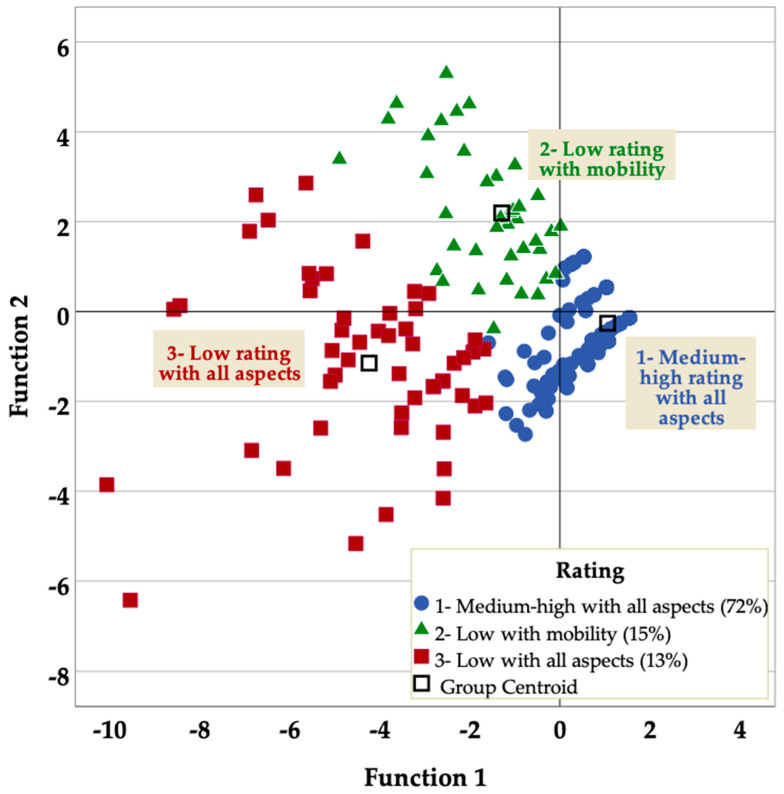
Grouping of subjects according to residential assessment. All-clusters scatter plot.

**Table 1 ijerph-19-16354-t001:** Assessment of nursing-home aspects during the pandemic period (in descending order based on the good/very good response category).

Aspects	Assessment (%)		N
	Well/Very Well	Badly/Very Badly	
10.How well their residential home has managed during the COVID-19 pandemic period	96.1	0.9	431
7. Their perception of safety in the residential home	95.9	0.7	434
2. Relationships with nursing-home workers	95.2	1.1	435
9. Having a space for privacy (having a single room or other personal space so as not to be disturbed)	93.8	2.8	435
3. Relationships with family members (e.g., visits in person, by phone, email, other means)	93.7	3.1	415
8. Characteristics and quality of the residential home’s amenities; (e.g., room size and design, natural lighting, noise level, temperature, green areas)	93.1	1.8	437
4. Relationships with friends (e.g., visits in person, by phone, email, other means)	89.2	3.7	323
1. Friendships with other residents	87.0	2.3	431
5. Getting around easily within the residential home (e.g., a lack of obstacles or barriers, ramps, stairs) (indoor accessibility)	85.5	5.9	422
6. The chance to go outside the residential home for different activities (e.g., walking around the neighbourhood or town, shopping, doing other leisure activities, visiting friends) (outdoor accessibility)	76.9	13.2	416

**Table 2 ijerph-19-16354-t002:** Logistic regression model for residential assessment cluster 1: medium-high rating for all aspects of the residential environment.

						95% C. I. for EXP(B)	
Variables in the Equation	Categories	B	Sig.	Exp(B)	Exp(B) Increment (%)	Lower	Upper
P.6. The Positive and Negative Affects Schedule (PANAS Balance)		0.08	0.011	1.08	8.15%	1.02	1.15
P.4. Fear of developing COVID-19, according to the FCV-19S scale		0.06	0.004	1.06	5.67%	1.02	1.10
P.16. Global Quality of Life (the FUMAT-24 scale adapted for elderly people without severe cognitive impairment in nursing homes)		0.04	0.050	1.04	3.85%	1.00	1.08
P14A- The Abbreviated Geriatric Depression Scale-Yesavage, 5 items		−0.29	0.039	0.75	−25.43%	0.56	0.99
P.9. Clusters of subjects according to leisure activities performed	(Reference: Inactivity cluster)		0.001				
	**1: Use of mobile devices**	**−0.15**	**0.828**	**0.86**		**0.23**	**3.23**
	2: Rewarding activities, volunteering	1.30	0.001	3.67	267.32%	1.67	8.09
	3: Religious practice	−1.14	0.033	0.32	−68.09%	0.11	0.91
P.10. Room availability	(Reference: Individual/private room)		0.032				
	**1: Room shared with my husband/wife/partner/other relative**	**−0.22**	**0.695**	**0.80**		**0.26**	**2.45**
	2: Room shared with other resident	−0.79	0.009	0.46	−54.48%	0.25	0.82
P.11. Where did you spend most of your time during the lockdown?	(Reference: Always in my room)						
	1: Limited use of different areas, unrestricted use of all spaces in care home	−0.78	0.016	0.46	−54.18%	0.24	0.87
	Constant	−2.39	0.13	0.09			
	Percentage of cases correctly classified: 77.5						
	R Square: Cox and Snell: 0.215; Nagelkerke: 0.311						
	Hosmer and Lemeshow test: chi-square: 5.592; df: 8; Sig: 0.693						
	In bold, non-significant categories						
	Variables entered in step: 1: Global Quality of Life (the FUMAT scale, 24 items); 2: Clusters of subjects according to leisure activities performed; 3: Where did you spend most of your time during the lockdown? 4: The Positive and Negative Affects Schedule (PANAS Balance); 5: Fear of developing COVID-19, according to the FCV-19S scale; 6: Room availability; 7: The Abbreviated Geriatric Depression Scale-Yesavage.						

**Table 3 ijerph-19-16354-t003:** Logistic regression model for residential assessment cluster 2: Low rating for mobility in the residential environment.

						95% C. I. for EXP(B)	
Variables in the Equation	Categories	B	Sig.	Exp(B)	Exp(B) Increment (%)	Lower	Upper
P.17. Devices or mobile systems used to maintain relationships with your family or friends (e.g., PC, tablet, telephone, mobile)	(Reference: Yes, through my own devices)		0.019				
	1: Yes, through other residents/staff’s devices	1.32	0.010	3.75	274.63%	1.37	10.25
	**2: No, I do not have access to those devices**	**−0.60**	**0.371**	**0.55**		**0.15**	**2.05**
P.22. Gender	(Reference: Male)						
	1: Female	1.18	0.004	3.25	224.57%	1.45	7.29
P.13. Self-perception of the stay in the nursing home in comparative perspective (before and during the COVID-19 pandemic)		1.04	0.000	2.82	182.13%	1.61	4.94
P14A- The Abbreviated Geriatric Depression Scale-Yesavage, 5 items		0.35	0.017	1.42	41.73%	1.06	1.89
P.9. Clusters of subjects according to leisure participation	(Reference: Inactivity cluster)		0.005				
	**1: Use of mobile devices**	**0.32**	**0.701**	**1.37**		**0.27**	**6.95**
	2: Rewarding activities, volunteering	−1.74	0.005	0.17	−82.53%	0.05	0.60
	**3: Religious practice**	**1.12**	**0.054**	**3.05**		**0.98**	**9.51**
	Constant	−5.49	0.000	0.00			
	Percentage correctly classified: 87.1						
	R Square: Cox and Snell: 0.166; Nagelkerke: 0.291						
	Hosmer and Lemeshow test: chi-square: 6.681; df: 7; Sig: 0.463						
	In bold, non-significant categories						
	Variables entered in step: 1: Self-perception of the stay in the nursing home in comparative perspective (before and during the COVID-19 pandemic) 2: Clusters of subjects according to leisure participation; 3: Gender; 4: Devices or mobile systems used to maintain relationships with your family or friends (e.g., PC, tablet, telephone, mobile); 5: The Abbreviated Geriatric Depression Scale-Yesavage.						

**Table 4 ijerph-19-16354-t004:** Logistic regression model for residential assessment cluster 3: Low rating for all aspects of the residential environment.

						95% C. I. for EXP(B)	
Variables in the Equation	Categories	B	Sig.	Exp(B)	Exp(B) Increment (%)	Lower	Upper
P.11. Where did they spend most of their time during the lockdown?	(Reference: Always in my room)						
	1: Limited use of different areas, unrestricted use of all spaces in care home	1.95	0.000	7.01	600.60%	3.12	15.75
P.23. Marital status	(Reference: Married/living with partner)		0.003				
	**1: Separated/Divorced**	**0.87**	**0.222**	**2.39**		**0.59**	**9.66**
	2: Single	1.84	0.001	6.29	528.74%	2.05	19.25
	3: Widow/widower	1.40	0.002	4.06	305.82%	1.69	9.77
P.3. Satisfaction with life in comparative perspective (before and during the COVID-19 pandemic)		1.20	0.001	3.33	233.18%	1.59	7.00
P.16. Quality of life as a whole (the FUMAT-24 scale adapted for elderly people without severe cognitive impairment in nursing homes)		−0.07	0.004	0.93	−6.75%	0.89	0.98
P.6. The Positive and Negative Affects Schedule (PANAS Balance)		−0.12	0.010	0.89	−11.04%	0.81	0.97
P.13. Self-perception of the stay in the nursing home in comparative perspective (before and during the COVID-19 pandemic)		−1.50	0.000	0.22	−77.75%	0.11	0.44
	Constant	3.18	0.102	24.16			
	Percentage of cases correctly classified: 91.0						
	R Square: Cox and Snell: 0.207; Nagelkerke: 0.394						
	Hosmer and Lemeshow test: chi-square: 6.760; df: 8; Sig: 0.563						
	In bold, non-significant categories						
	Variables entered in step: 1: Where did they spend most of their time during the lockdown? 2: Self-perception of the stay in the nursing home in comparative perspective (before and during the COVID-19 pandemic); 3: Global Quality of Life (the FUMAT scale, 24 items); 4: Marital status; 5: Satisfaction with life in comparative perspective (before and during the COVID-19 pandemic); 6: The Positive and Negative Affects Schedule (PANAS Balance).						

## Data Availability

Not applicable.

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
