# Peer review of "Residential Environment Assessment by Older Adults in Nursing Homes during COVID-19 Outbreak"

_ijerph, 2022, doi:10.3390/ijerph192316354_

Round 1

Reviewer 1 Report

Very well written paper - based on highly relevant data. As for further discussion & future research the authors might also consider the extensive work burden on the nursing home empolyees due to the pandemic. An additional study on that issue would help to complete a holistic view on the environment "nursing homes for the elderly".  

Author Response

The authors are very grateful for the support provided by the reviewer to enhance the manuscript.

Reviewer pointed out that the manuscript was “Very well written paper”. Even though, the specialized translator, who translated the original document into English, has revisited the text, and has changed a number of things, as follows:

Line 59, “place os residence” has been changed by “place of residence”

Line 135: “stdies” has been changed by “studies”

Table 1: “eg” has been changed by “e.g.”.

Table 3, foot note: “eg” has been changed by “e.g.”.

Table S1: “eg” has been changed by “e.g.”.

Please, note that all changes are showed in yellow color in the manuscript.

As expressed in the last section of the manuscript, we are in the process of completing this research from a double perspective: quantitative and longitudinal, in one hand, and considering all contexts of this residential strategy (residents, workers, family) based in a qualitative approach, in the other hand.

Our thanks to the reviewer.

Reviewer 2 Report

This is a complex paper with some very detailed statistical analysis.  The goals of the paper are very broad and the last of these (lines 146-7) is not very specific. 

The methodology and the presentation of the results are very complex (especially to a reader without advanced statistical skills), therefore, I was expecting the discussion to provide a clearer illustration of some of the main findings.

The conclusion could also have made more effort to explain the implications for public policy.  The main conclusions seem to be that private rooms with high quality amenities, friendly and professional staff and good means of communication with family and friends have an important impact on well-being and, not surprisingly, this was very apparent during the Covid-19 pandemic.

The English could possibly simplified and ther is a spelling error on line 134 (stdies).

Author Response

The authors are very grateful for the support provided by the reviewer in his/her task in order to enhance it.

Following, we answer the reviewer according to his/her comments and suggestions.

1) Reviewer comment: “This is a complex paper with some very detailed statistical analysis. The goals of the paper are very broad and the last of these (lines 146-7) is not very specific”.

Authors answer: the authors stated the objectives to maximize their clarification. However, following the reviewer suggestion and in order to achieve greater clarification, authors have changed the expression of the objectives appeared in lines 146-149, as: “iii) to group subjects into clusters of similar characteristics according to residential assessment; iv) to find out the personal and contextual facts that condition the position of the subjects in each of the residential assessment clusters.”.

2) Reviewer comment: “The methodology and the presentation of the results are very complex (especially to a reader without advanced statistical skills), therefore, I was expecting the discussion to provide a clearer illustration of some of the main findings.”.

Authors answer: Social reality is complex, which is why the authors paid special attention in the study design stage, its approach and results report. We consider that the description of the results has been carried out in the manner in which these types of analyses are usually presented in scientific documents.

In this sense, the authors have been especially careful in expressing the multivariate analysis results. In fact, in the Material and Methods section, subsection 2.2. Statistical analysis, in addition to describing the statistical techniques used, the purpose of their use was detailed in an accessible way.

Likewise, the results of some multivariate techniques have been presented as supplementary material, just to avoid a possible reader misunderstanding when confronted with multivariate results. However, these results have been conveniently described in an easy manner in the corresponding section. In the same way, and for a better comprehension, the results of the regression models have been explained in text as well straightforward, in the way in which these types of regression data are usually presented in scientific literature.

In addition, it is noteworthy that the results have been conveniently argued in section 4. Discussion.

For all of which, and in line with the reviewer's comments, the authors have re-read the document, and now we believe that it is more understandable.

3) Reviewer comment: “The conclusion could also have made more effort to explain the implications for public policy. The main conclusions seem to be that private rooms with high quality amenities, friendly and professional staff and good means of communication with family and friends have an important impact on well-being and, not surprisingly, this was very apparent during the Covid-19 pandemic.”

Authors answer:

This is a relevant observation of the reviewer. The purpose of this study is in line with the advancement of scientific knowledge. However, the text in lines 589-603 points out or synthesizes the future needs for the design of residential spaces for old people in nursing homes based on the results. We believe that this could be enough for a manuscript whose object is research, and not the implications for public policies, a matter of enormous relevance for policy-makers and other experts in the design and implementation of public policies. In any case, and in light of the ongoing study from a double methodological approach, we consider it is relevant to delve into this line, which is of enormous interest to transfer the research results to society.

4) Reviewer comment: The English could possibly simplified and ther is a spelling error on line 134 (stdies).

Authors answer: In order to clarify this point, the specialized translator, who translated the original document into English, has revisited the text, and has changed a number of things, as follows:

Line 59, “place os residence” has been changed by “place of residence”

Line 135: “stdies” has been changed by “studies”

Table 1: “eg” has been changed by “e.g.”.

Table 3, foot note: “eg” has been changed by “e.g.”.

Table S1: “eg” has been changed by “e.g.”.

Please, note that all changes are showed in yellow color in the manuscript.

Finally, we are very grateful to the reviewer's considerations in order to enhance the paper, and we remain at his/her disposal.